# Trends of Korean Medicine Treatment for Parkinson’s Disease in South Korea: A Cross-Sectional Analysis Using the Health Insurance Review and Assessment Service–National Patient Sample Database

**DOI:** 10.3390/healthcare13101207

**Published:** 2025-05-21

**Authors:** BackJun Kim, Huijun Kim, Ye-Seul Lee, Yoon Jae Lee, In Chul Jung, Ju Yeon Kim, In-Hyuk Ha

**Affiliations:** 1Jaseng Hospital of Korean Medicine, 536 Gangnam-daero, Gangnam-gu, Seoul 06110, Republic of Korea; junant109@naver.com; 2Jaseng Spine and Joint Research Institute, Jaseng Medical Foundation, 540 Gangnam-daero, Gangnam-gu, Seoul 06110, Republic of Korea; khj26677@gmail.com (H.K.); yeseulee@jaseng.org (Y.-S.L.); goodsmile@jaseng.org (Y.J.L.); 3Department of Oriental Neuropsychiatry, College of Korean Medicine, Daejeon University, Daejeon 34520, Republic of Korea; npjeong@dju.kr

**Keywords:** acupuncture, dopamine, insurance, musculoskeletal system, Parkinson’s disease, Republic of Korea

## Abstract

**Background/Objectives:** Parkinson’s disease (PD) is a major neurodegenerative condition, mainly treated using dopamine-based therapies. However, the side effects and limitations of these therapies hinder their use. This study aimed to analyze the utilization of Korean medicine (KM) by patients with PD in Korea. **Methods**: Data of the Health Insurance Review and Assessment Service–National Patient Sample were used to investigate the status and trend of KM utilization by patients with PD in Korea from January 2010 to December 2019. Data of 18,562 patients were included, and analyses were performed on the status of KM and Western medicine (WM) utilization, cost of care, prescribed medications, comorbidities, and characteristics of patients with PD. **Results**: The number of patients who utilized KM services for PD gradually increased over the 10-year period, with 10.6% of all patients with PD using KM services in 2019. In addition, the number of KM users with PD, number of claims, and expenses all showed an increase. The rate of increase in KM service utilization was greater than that of WM. Among KM services, acupuncture had the highest expense (50.6%). Regarding comorbidities in patients with PD, musculoskeletal diseases were the most common (58.6%). Among WM medications prescribed for the KM users, dopa and dopa derivatives (15.5%) and anti-dementia drugs (11.7%) were the most common. **Conclusions**: This study provides useful information on KM utilization status and trends among patients with PD and the characteristics of these patients. Follow-up research is warranted on the utilization status of more diverse KM services.

## 1. Introduction

Parkinson’s disease (PD) is a progressive neurodegenerative disorder characterized by the loss of dopaminergic neurons in the substantia nigra of the brain. The typical symptoms of PD comprise a tetrad of motor deficits, including resting tremor, bradykinesia, postural instability, and rigidity of the neck, trunk, and limbs [1]. These symptoms arise from basal ganglia damage, causing dysfunction in procedural learning [2] or a multi-system disorder with neuroinflammation and immune dysfunction [1].

The prevalence of PD is increasing worldwide. In 2016, approximately 6.1 million individuals were diagnosed with PD globally, which is 2.4 times higher than that in 1990 [3]. In South Korea, the prevalence of PD per 100,000 population was 41.4 in 2004 and 142.5 in 2013, showing an annual increase of 13.2% over the 10-year period. In 2017, the number of patients who utilized medical services for PD treatment accounted for 0.21% of all patients in Korea [4].

Pharmacologic treatments for PD in conventional medicine (hereafter referred to as Western medicine [WM]) are mainly dopamine-based therapies using dopamine replacement agents. The currently available treatments include levodopa, dopamine agonists, and monoamine oxidase-B inhibitors. Furthermore, deep brain stimulation is used as a surgical treatment [5]. However, none of these methods are recognized as disease-modifying treatments. In addition, patients with PD exhibit not only motor symptoms but also non-motor symptoms, such as cognitive dysfunction/impairment, depression, insomnia, hallucinosis, and autonomic dysfunction, causing difficulties in their activities of daily living and a decline in their quality of life [6]. Dopamine replacement therapies show minimal to no improvement in the non-motor symptoms [2].

With the progression of PD and the limited efficacy of WM treatments, Korean patients utilize traditional Korean medicine (KM). According to Health Insurance Review and Assessment Service (HIRA) statistics [7] in Korea, PD ranked 16th among the diagnosis codes of inpatients utilizing KM in 2021. A previous study [8] reported that approximately 76% of 123 patients with PD used complementary and alternative medicine (CAM) services, such as KM, to improve PD symptoms, including motor symptoms (57.6%), fatigue (19.6%), constipation (5.4%), and pain (4.3%). In the United States, approximately 40% of patients with PD were reported to have used CAM services at least once [9]. A multinational review reported that the prevalence of CAM use among patients with PD ranged from 25.7% to 76% worldwide, with frequently utilized therapies including acupuncture, herbal medicine, massage, and supplements. These therapies were mainly used to improve motor symptoms, and the widespread adoption of CAM for PD has been confirmed in multiple countries. In recent years, clinical research and national initiatives in Korea have increasingly focused on providing evidence-based KM, including the development of clinical practice guidelines for PD. Therefore, the present study on KM utilization in patients with PD may serve as a valuable reference for expanding global CAM usage and informing international health policy [10].

Despite the high demand and actual utilization of KM services for PD treatment, research evidence or nationwide data reflecting the recent trend of KM service utilization for PD treatment in Korea remain scarce. The majority of existing studies on PD are experimental [11], and the few clinical studies are mainly retrospective studies or case reports. Furthermore, the study period has tended to be short, or studies have only covered patients visiting a single KM hospital [12,13,14], resulting in difficulties in generalizing the findings to understand the overall status and trend of KM healthcare utilization. Thus, there is a clear need for analysis and reporting on the use of KM services for PD treatment based on HIRA data.

Therefore, this study aimed to investigate the status of the provision of WM and KM services and their utilization for PD treatment, medical costs associated with PD treatment, comorbidities, and patient characteristics using 10-year (2010–2019) National Health Insurance (NHI) claims data provided by HIRA.

## 2. Materials and Methods

### 2.1. Study Design and Data Source

This study used HIRA–National Patient Sample (NPS) data from January 2010 to December 2019. The data comprised claims generated when healthcare providers requested reimbursement/payment from NHIS. The HIRA–NPS data comprise (1) social demographic characteristics of patients with PD (age, sex, health insurance type, etc.); (2) general information on healthcare utilization for patients with PD (outpatient/inpatient status, medical department, type of medical institution, total number of days receiving medical care or using medical services, length of hospital stay for inpatients/period of service for outpatients, etc.); (3) KM service utilization for patients with PD (KM services based on the standards presented in the Notice of the Ministry of Health and Welfare, fees for the services provided, and diagnosis). HIRA–NPS consists of a 3% sample (approximately 1,375,842 people for 2011) from sex-stratified (two strata) and age-stratified (16 strata) random sample data from the entire Korean population each year. Since the NHI provides health insurance coverage for 98% of the Korean population, the HIRA–NPS data adequately represent the entire patient population in Korea and serve as a useful source of information for healthcare research [15,16]. All statistical analyses were performed using SAS 9.4 (SAS Institute Inc., Cary, NC, USA).

### 2.2. Study Population

This study utilized 10-year (2010–2019) patient sample data from the HIRA–NPS. The 2010–2018 data had a patient sampling ratio of 3% of the total population, and data for the year 2019 had a patient sampling ratio of 2%. Based on the claims data for WM and KM services, patients with at least one record of treatment for PD (G20; Parkinson’s disease), as the primary diagnosis, were included in the study.

#### Eligibility Criteria

The inclusion criteria were male and female patients of all age groups who used medical services for PD (International Classification of Diseases, 10th revision [ICD-10] code: G20) as the primary diagnosis. The exclusion criteria were records of diagnosis with G21 (secondary parkinsonism), G22 (parkinsonism in diseases classified elsewhere), or G23 (other degenerative diseases of basal ganglia) as the primary or secondary diagnosis among patients with PD (G20) as the primary diagnosis. The following data were excluded from the final analysis: claim billing code that did not represent WM or KM institutions; code corresponding to mental hospitals, dental hospitals and clinics, midwife service centers (birthing centers), or pharmacies or missing code; missing records for the medical department code or missing amount of benefit covered by NHI.

### 2.3. Calculation Methods

To ensure accessibility and convenience in utilizing data, HIRA provides sample data in 1-year segments extracted via random stratification. The raw data were first deidentified by removing any information that could reveal the identity of individuals or entities/corporations. Secondary data were then generated through statistical sampling, and the final dataset was built using records of treatment that had been claimed for 1 year based on the start date of medical care for each applicable year. For all patients who used medical services for 1 year, patient-level stratified systematic sampling was performed according to the sex and age strata (in 10-year increments), including treatment and prescription records.

### 2.4. Statistical Analysis

Descriptive statistics were performed in this study. For sociodemographic characteristics and healthcare utilization by patients with PD, categorical variables are expressed as frequency (N) and percentage (%), whereas continuous variables are expressed as mean, standard deviation, minimum value, and maximum value. For a more in-depth understanding of the status and trends of healthcare utilization, analyses were performed considering age, medical department, and type of medical institution. All analyses were performed using the statistical software package SAS V9.4 (SAS Institute Inc., Cary, NC, USA).

### 2.5. Ethics Approval

The current study was reviewed and qualified for an exemption for ethical approval by the Institutional Review Board of Jaseng Hospital of Korean Medicine, Seoul, Korea (ethical code: 2023-07-017, 10 July 2023).

## 3. Results

### 3.1. Patient Enrollment

Figure 1 illustrates the process of identifying patients with a primary diagnosis of PD from the sample data, applying the eligibility criteria, and determining the final study population. From 2010 to 2019, 19,909 patients were diagnosed with PD (ICD-10 code: G20) as the primary diagnosis. Of these, 12 patients were excluded owing to a billing code that did not represent WM or KM institutions or because of missing data on the medical department code, type of healthcare institution code, or total amount of benefit to be covered by NHI. Furthermore, 1335 patients who had been diagnosed with atypical or secondary parkinsonism as the primary or secondary diagnosis among those with idiopathic PD (ICD-10 code: G20) were excluded. Therefore, a total of 18,562 patients were included in the analysis (Figure 1).

### 3.2. Basic Patient Characteristics

Patients were categorized into two groups: those who visited WM institutions only (those who used WM services only: non-KM users) and those who visited KM institutions at least once (those who used KM services: KM users).

Among KM users, there were more female (64.9%) than male patients (35.1%). Regarding the group distribution, those aged ≥ 70 years were the majority (65.6%). Regarding health insurance type, NHI accounted for the highest proportion (90.0%), and Medicaid comprised 9.9% of the patients. The trends of sex ratio and health insurance type among KM users showed no major differences compared with the trends among non-KM users (Table 1).

### 3.3. Basic Medical Usage Characteristics

Of the total number of patients with PD, 9.7% visited KM institutions (KM hospitals and KM clinics) at least once, and 90.2% visited only WM institutions (hospitals and clinics). Among claims from visits to medical institutions for PD treatment, the proportion of outpatient services was 79.7%, and that of inpatient services was 20.3% (Table 2).

### 3.4. Annual Trends in the Total Number of Patients with PD Who Used Medical Services and the Ratio of KM Users

For the 10-year period from 2010 to 2019, the number of patients who visited either WM or KM institutions for PD treatment showed an increasing trend from 1333 patients in 2010 to 2260 patients in 2019. The ratio of KM users to the total patient population with PD increased from 6.4% in 2010 to 10.6% in 2019. In contrast, the ratio of non-KM users to the total patient population with PD decreased from 93.6% in 2010 to 89.4% in 2019 (Appendix A).

### 3.5. Annual Trends in the Number of Claims and Costs for Healthcare Utilization

For the 10-year period, the number of KM users increased by approximately 181.2% (from 85 patients in 2010 to 239 patients in 2019) (Figure 2a), whereas the number of non-KM users increased by 61.9% (from 1248 in 2010 to 2021 in 2019).

The number of claims for KM services showed an overall increasing trend, rising by approximately 386.5% from 959 in 2010 to 4666 in 2019 (Figure 2b). For the same period, the claims for WM services by non-KM users increased by 68.7% (from 8011 in 2010 to 13,520 in 2019).

The total annual expense for the KM users showed an overall increasing trend, rising by approximately 815.3% from USD 199,981 in 2010 to USD 1,830,368 in 2019 (Figure 2c). The total annual expenses for the non-KM users increased by 156.2% (from USD 2,486,123 in 2010 to USD 6,369,601 in 2019). Thus, the rate of increase in the total medical expenses for the KM users was greater than that for the non-KM users.

The annual expense per patient among KM users also showed an overall increasing trend, rising by approximately 225.5% from USD 2353 in 2010 to USD 7658 in 2019 (Figure 2d). The annual expense per patient among non-KM users increased by 58.2% (from USD 1992 in 2010 to USD 3152 in 2019), indicating that the rate of increase in the expense per patient was larger for KM users than for non-KM users.

### 3.6. Analysis of PD-Related KM Services Covered by NHI, Number of Claims, and Costs

The PD-related KM services covered by NHI were categorized as follows: acupuncture, consultation, physical therapy, moxibustion, cupping, electroacupuncture, dispensing, meal service fee, admission, rehabilitation assistive devices, Chuna, examination, procedure, psychotherapy, and others. For the 10-year period, the number of claims was the largest in acupuncture (18,806 cases) and consultation (18,753 cases), accounting for 28.8%, respectively. Regarding total costs, acupuncture had the highest cost (USD 376,008, 50.6%), followed by consultation (USD 138,398, 18.6%), cupping (USD 67,359, 9.1%), and moxibustion (USD 54,460, 7.3%) (Appendix A).

The top 10 KM services covered by NHI, excluding PD-related admission and consultation, were analyzed. Acupuncture (two regions) accounted for the largest number of claims (14,703 cases, 21.7%), followed by perforating acupuncture (11,728 cases, 17.3%), transcutaneous infrared irradiation therapy (7184 cases, 10.6%), and electroacupuncture (4666 cases, 6.9%). Regarding total costs, general acupuncture (with acupoints in two or more regions) was the most expensive (USD 177,227, 32.2%), followed by perforating acupuncture (USD 143,606, 26.1%) and dry cupping therapy (USD 45,951, 8.3%) (Appendix A).

### 3.7. Annual Trends in the Number of Claims and Costs for Major PD-Related KM Services Covered by NHI

For the 10-year period from 2010 to 2019, acupuncture claims accounted for the highest proportion, and this trend remained similar throughout the 10-year period. Physical therapy accounted for the second highest number of claims for the 10-year period, except in 2011 and 2012 (Figure 3). Annual trends in medical costs for each KM service showed trends similar to the trends in the number of claims (Appendix A).

### 3.8. Frequently Diagnosed Diseases in Claims for PD-Related KM Services

The frequencies of diagnoses shown in the claims for PD-related KM services were analyzed based on the first digit of the primary diagnosis. Diagnoses of M-code (diseases of the musculoskeletal system and connective tissue; M00–M99) were the most frequent (78,318 cases, 58.6%), followed by those of U-code (codes for special purposes; U00–U99) (14,640 cases, 11.0%), S-code (injury, poisoning, and certain other consequences of external causes; S00–T98) (12,235 cases, 9.2%), and R-code (symptoms, signs, and abnormal clinical and laboratory findings, not elsewhere classified; R00–R99) (8776 cases, 6.6%) (Table 3).

For more detailed analyses, the frequencies of diagnoses shown in the claims for PD-related KM services were analyzed based on the first three digits of the primary diagnosis. Dorsalgia (M54) was the most frequently diagnosed disease (37,478 cases, 28.1%), followed by other and unspecified soft tissue disorders, not elsewhere classified (M79) (11,124 cases, 8.3%), knee osteoarthritis (M17) (6954 cases, 5.2%), shoulder lesions (M75) (5441 cases, 4.1%), dislocation and sprain of joints and ligaments of lumbar spine and pelvis (S33) (4728 cases, 3.5%), and other joint disorders not elsewhere classified (M25) (4691 cases, 3.5%) (Appendix A).

### 3.9. Analysis of WM Prescription Details for Patients with PD

Information on WM prescription (i.e., medications prescribed by WM services) was analyzed for KM users. The drugs were largely categorized into anti-Parkinson drugs and others based on their generic names and the Anatomical Therapeutic Chemical Classification System. The anti-Parkinson drugs and the top 10 most frequently prescribed drugs are summarized in Table 4.

Based on the total number of claims, in the category of anti-Parkinson drugs, dopa and dopa derivatives were the most frequently prescribed (5365 cases, 15.2%), followed by dopamine agonists (3368 cases, 9.5%), amantadine derivatives (1359 cases, 3.8%), monoamine oxidase B inhibitors (911 cases, 2.6%), and catechol-o-methyl-transferase (COMT) inhibitors (114 cases, 0.3%). In the category of “others,” drugs for the alimentary tract and metabolism were the most frequently prescribed (4082 cases, 11.5%), followed by anti-dementia drugs (4025 cases, 11.4%) and the cardiovascular system (3028 cases, 8.6%) (Table 4).

### 3.10. Subgroup Analysis

#### 3.10.1. Demographic Characteristics According to KM Utilization Frequency

Among patients who used KM at least once, we conducted a subgroup analysis of basic demographic characteristics based on a 20-visit threshold. The results showed that patients who used KM more frequently (≥20 visits) tended to include a higher proportion of males (45.8%) and were more concentrated in the 50s (15.1%) to 60s (33.6%) age group compared to those who used KM less frequently (<20 visits) (Table 5).

#### 3.10.2. Annual Trends and Patient Characteristics by KM Utilization and Anti-Dementia Drug Prescription

A subgroup analysis was conducted to examine KM utilization among patients prescribed anti-dementia drugs. When analyzing the annual trend, a consistent increase was observed in the number of patients from 2010 to 2019 (Table 6).

Regarding the basic characteristics, compared to the overall study population, the proportion of females was higher (68.3%), and the majority of patients (78.6%) were aged 70 years or older (Table 7).

## 4. Discussion

This study used HIRA claims data to investigate the provision and utilization of KM services for patients with idiopathic PD in Korea over a 10-year period from 2010 to 2019. We observed that the majority of patients with PD used WM services (18,208 out of 18,562 patients) as the primary treatment approach, with only a few patients utilizing KM services as additional medical care. The characteristics of patients did not differ between the KM and non-KM users. The proportions of males and females among patients with PD were 39.3% and 60.7%, respectively, indicating more female patients with PD. According to age, the majority of patients with PD (89.9%) were >60 years old. In 2013, the prevalence of PD among those aged ≥ 60 years in Korea was 716.0 per 100,000 population, which was approximately 5-fold higher than the prevalence for the total population (142.5 per 100,000 population). A previous study reported that the number of male patients with PD increased annually by approximately 12.8%, from 35.4 per 100,000 in 2004 to 117.7 per 100,000 in 2013, and the number of female patients increased annually by approximately 13%, from 47.4 in 2004 to 167.3 in 2013. These results show a similar level of annual average increase in male and female patients, with more female than male patients during a 10-year period [6].

A previous study [17] showed that the 70- to 79-year age group accounted for 34.10% of patients with PD, the 80- to 89-year age group accounted for 22.79%, and the ≥90-year age group accounted for 3.55%. This result is consistent with the findings of a previous study [18], in which the incidence of PD per 100,000 population in Asia was highest among those aged 70–79 years. Furthermore, in the previous study, the number of female patients was approximately 1.5 times higher than that of male patients, which is similar to the male-to-female ratio (0.7–2.4) reported worldwide [18].

The total number of patients with PD increased from 1333 in 2010 to 2260 in 2019. For the 10-year period, the number of KM users increased by 181.2%, the number of claims for KM services increased by 386.5%, the total annual expenses for KM services increased by 815.3%, while non-KM users showed a 156.2% increase during the same period. This disparity was also evident in per-patient annual expenses, which rose by 225.5% for KM users and by 58.2% for non-KM users. These rates of increase were all greater than the rates of increase in the same categories among non-KM users. In 2010, only 6.4% of the patients with PD used KM services; however, since 2017, >10% of the patients have used KM services. Thus, these results indicate that the demand for KM services by patients with PD has gradually increased over time.

Several factors may contribute to this trend. First, the procedural characteristics of KM services, such as acupuncture—which accounted for 50.6% of KM costs—may lead to higher cumulative costs compared to the pharmacological focus of non-KM services. Second, the demographic profile of KM users, with 65.6% aged 70 years or older and a high prevalence of musculoskeletal comorbidities (58.6% M-code diagnoses), likely increases the need for multimodal and repeated interventions. Third, while NHI covers KM physical therapies, it does not reimburse herbal medicines, which are used by a significant proportion of patients, potentially underestimating the true economic burden on KM users. These findings reflect the broader context of an aging population in Korea and increased interest in complementary and alternative medicine for chronic neurodegenerative diseases. Therefore, future policy should consider the cost-effectiveness and accessibility of KM services when planning integrated care for PD.

These findings highlight the importance of KM in addressing not only motor but also non-motor symptoms of PD, such as pain, dyspepsia, and dizziness. Notably, our data showed that musculoskeletal pain, functional dyspepsia, and dizziness were more frequently managed with KM compared to general KM users. This reflects the clinical demand for integrative approaches to symptom control in PD. Recent clinical practice guidelines recommend acupuncture and herbal medicine—including formulas such as Cheonmagudeungeum and Sukjipyeongjeontang—for non-motor symptoms, including gastrointestinal dysfunction and dizziness, based on accumulated clinical evidence and expert consensus [19]. Furthermore, systematic reviews and real-world observational studies have demonstrated that herbal medicine and acupuncture can improve gastrointestinal symptoms, pain, and quality of life in PD patients, with a favorable safety profile, although a moderate to high risk of bias was present among the included studies [20,21]. Most systematic reviews on acupuncture for PD included randomized controlled trials, but the methodological quality of the included studies was generally low to moderate. These results support the need to promote evidence-based KM interventions for non-motor symptoms in clinical practice and public health policy, including insurance coverage expansion and multidisciplinary care models.

Several mechanisms have been proposed to explain the efficacy of acupuncture in Parkinson’s disease. Systematic reviews have indicated that acupuncture may enhance dopaminergic neurotransmission and modulate basal ganglia circuits, thereby contributing to symptom relief [22]. Additional meta-analyses have reported that acupuncture can regulate neurotransmitter balance, reduce neuroinflammation, and improve both motor and non-motor symptoms in PD patients [23]. Furthermore, functional neuroimaging studies have demonstrated that acupuncture activates motor-related brain regions, such as the putamen and primary motor cortex, supporting its clinical benefits for PD [24]. These multimodal mechanisms—including neurotransmitter regulation, anti-inflammatory action, and neuroprotection—are considered to underlie the observed improvements following acupuncture treatment [22,23,24].

In terms of the number of claims for each KM service, general acupuncture (two regions) had the largest (14,703 cases, 21.7%), followed by perforating acupuncture (11,728 cases, 17.3%), transcutaneous infrared irradiation therapy (7184 cases, 10.6%), electroacupuncture (4666 cases, 6.9%), and intra-articular acupuncture (4091 cases, 6.0%). From 2010 to 2019, acupuncture was the most frequently used PD-related KM service covered by NHI, suggesting that acupuncture accounted for a significant proportion of the KM services. In terms of total cost, items related to acupuncture accounted for the highest cost.

Acupuncture has demonstrated its efficacy against PD symptoms in several studies. Among patients with PD, acupuncture resulted in a significant effect on nocturia, dysphagia, constipation, senile tremor, sleep disturbances, and fatigue [22]. In several systematic reviews, acupuncture as a monotherapy or in combination with levodopa showed superior efficacy in improving PD symptoms compared with levodopa treatment alone [22,23,24]. Scalp acupuncture is reported to improve cerebral vascular oxygen circulation and metabolism, cognitive function, mental health, and emotional function [24]. In addition, a combination of acupuncture and herbal medicine significantly reduced the King’s Parkinson’s disease Pain Scale (KPPS) total score and the score in the area of radicular pain [25]. Furthermore, combining acupuncture with bee venom acupuncture resulted in a significant improvement in gait speed, Parkinson’s Disease Quality of Life score, activities of daily living (Unified Parkinson’s Disease Rating Scale [UPDRS] part II), and motor symptoms (UPDRS part III) [26].

The results on comorbidities among KM users showed that the M and S codes, which are related to the diseases of the musculoskeletal system and connective tissue, accounted for the highest number of claims. A similar trend was observed in the 2021 healthcare costs statistical indices of Korea [27], in which the M and S codes were the top frequent claims associated with KM utilization. Patients with PD mainly used KM services for the management of musculoskeletal pain. Similarly, in a previous retrospective study investigating the types of pain in PD, 69.3% were from the musculoskeletal system, 39.4% were radicular and neuropathic pain, and 32.0% were from dystonia [25]. Patients with PD have a lower pain threshold than healthy individuals and are likely to develop muscle rigidity. In addition, motor symptoms, such as gait difficulty, postural instability, and tremor, impair balance, resulting in frequent and numerous complaints of pain in the musculoskeletal system [25].

Most patients with PD experience non-motor symptoms, which occur more frequently as the disease progresses [27]. According to the PRIAMO study [28], a large-scale multi-center study, 98.6% of patients with PD experienced non-motor symptoms, and the average number of non-motor symptoms per patient was 7.8. Furthermore, patients with PD have a high rate of gastrointestinal comorbidities, such as gastritis, dyspepsia, and constipation [8]. KM services have been reported to be effective in improving the non-motor symptoms [12], and patients also show preferences for KM services.

Notably, functional dyspepsia (K30) and dizziness and giddiness (R42) ranked 13th and 17th among comorbidities in patients with PD, higher than the respective ranks of 22nd and 42nd for KM users in general. These results indicated a higher use of KM services by patients with PD not only for pain, but also for dyspepsia and dizziness, compared with general KM users.

Regarding WM prescription of patients with PD, the frequently prescribed anti-Parkinson drugs were dopa and dopa derivatives, followed by dopamine agonists, amantadine derivatives, monoamine oxidase B inhibitors, and COMT inhibitors. Among the drugs classified as “Others”, excluding the drugs for the alimentary tract, metabolism, anti-dementia drugs were the most prescribed, followed by cardiovascular system drugs, and anxiolytics. The prevalence of mild cognitive impairment tends to increase with the progression of PD [29]; therefore, it can be inferred that PD had progressed to a certain stage among many patients using KM services, resulting in cognitive and functional impairment.

Subgroup analyses were conducted based on the additional insights into KM utilization patterns among patients with PD. Patients who received KM treatment more frequently (≥20 visits) tended to be male and were more commonly in their 50s and 60s, suggesting that KM services may be actively sought by younger, functionally active PD patients who are still engaged in daily or occupational activities. In contrast, among patients prescribed anti-dementia drugs, KM users were predominantly female and over 70 years of age, with a steady increase in patient numbers observed annually from 2010 to 2019. These findings reflect potential differences in healthcare-seeking behavior and symptom burden depending on age and cognitive status.

This study has several limitations. First, since this study was based on HIRA claims data, non-covered items in KM services, such as herbal medicine, were excluded from the analysis. However, patients receiving KM services for PD utilize herbal medicine more often than acupuncture. Among patients with PD who used CAM in a previous study, the most frequent treatment was herbal medicine (32.3%), followed by acupuncture (22.6%) [30]. In clinical practices in Korea, herbal medicines, such as Sukjipyeongjeontang and Cheonmagudeungeum [5], are prescribed for PD based on the conditions and identified patterns of individual patients. Various herbal formulations, such as Eokgansan, Yeoldahansotang, Cheongsimyeonjatang, and Hyangsayangyitang, were reported to be effective in reducing UPDRS II and III scores, postural instability and gait disorder, and bradykinesia/tremor scores, as well as improving nonmotor symptoms [12]. In addition, pharmacopuncture is utilized for PD treatment; Jungsongouhyul pharmacopuncture and bee venom acupuncture were reported to reduce the KPPS total score and pain. In particular, bee venom acupuncture has neuroprotective and anti-inflammatory effects [13,31,32]. Therefore, additional analysis on non-covered items of KM services is needed to improve the current understanding of KM utilization for PD treatment.

Second, to more accurately analyze KM utilization for idiopathic Parkinson’s disease, we included only patients with a primary diagnosis of G20. However, this approach may have underestimated KM use among patients who sought care for other chief complaints but also had comorbid Parkinson’s disease.

Third, in Korea’s dual healthcare system, KM and WM services are typically provided independently, operating under a concurrent model of care. Although integrative approaches are clinically important, there is a lack of studies examining cooperative care models using health insurance claims data. Future research is needed to explore the patterns and outcomes of integrative KM–WM care for Parkinson’s disease.

Fourth, as this study used HIRA claims data, it is unknown whether symptoms or the level of satisfaction with KM services improved. Dopamine replacement therapies are known to improve symptoms for the first few years; however, the efficacy tends to decrease over time, and side effects such as dyskinesia develop [33,34]. These complications occur in 50% of patients with PD after 5 years and in 80% of patients after 10 years of levodopa treatment. Given the limitations of WM treatment, the satisfaction and prognosis of KM users are useful research themes. Recent clinical guidelines and meta-analyses have shown that acupuncture, cupping, and integrative Korean medicine therapies are widely used for Parkinson’s disease in Korea, with evidence supporting improvements in both motor and non-motor symptoms [19,22,23]. Therefore, further research including patient-reported outcomes for patients with PD is needed. Studies linking elderly cohort data with hospital-based clinical records could enable more in-depth analyses of the status of healthcare utilization and therapeutic effects of a wide range of KM services, including non-covered interventions.

## 5. Conclusions

In summary, this study analyzed the trends and characteristics of Korean medicine (KM) utilization among patients with Parkinson’s disease (PD) in South Korea over a 10-year period. The findings revealed a steady increase in KM usage, particularly acupuncture, among PD patients, with musculoskeletal comorbidities being the most common. These results suggest that KM, as a complementary approach, plays an important role in addressing both motor and non-motor symptoms of PD. However, the study was limited by the exclusion of non-insurance-covered KM services, such as herbal medicine. Future research should include a broader range of KM interventions and evaluate patient-reported outcomes to better understand the effectiveness and satisfaction with KM in PD management. This research provides valuable insights for healthcare policy and the integration of KM in chronic neurodegenerative disease care in Korea.

## Figures and Tables

**Figure 1 healthcare-13-01207-f001:**
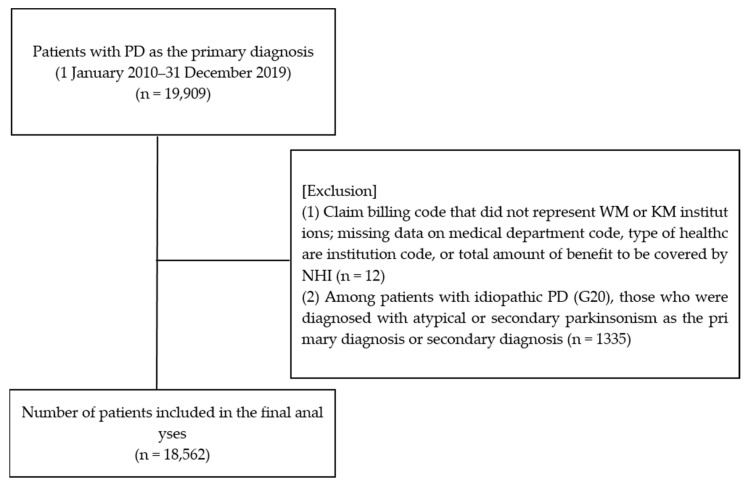
A flowchart for patient enrollment. KM: Korean medicine; NHI: National Health Insurance; PD: Parkinson’s disease; WM: Western medicine.

**Figure 2 healthcare-13-01207-f002:**
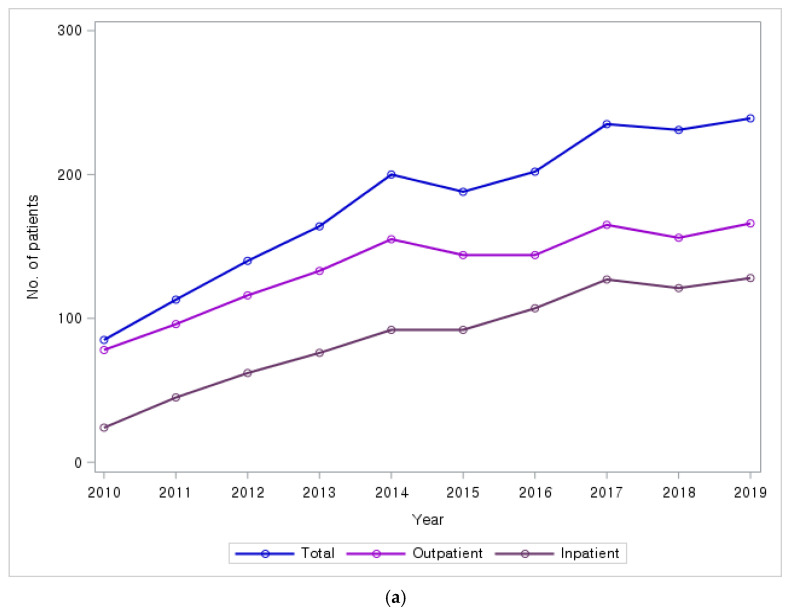
(**a**) Number of patients who used KM services at least once; (**b**) number of claims for using KM services; (**c**) total annual expenses for patients who used KM services at least once; and (**d**) annual expenses per patient for those who used KM services.

**Figure 3 healthcare-13-01207-f003:**
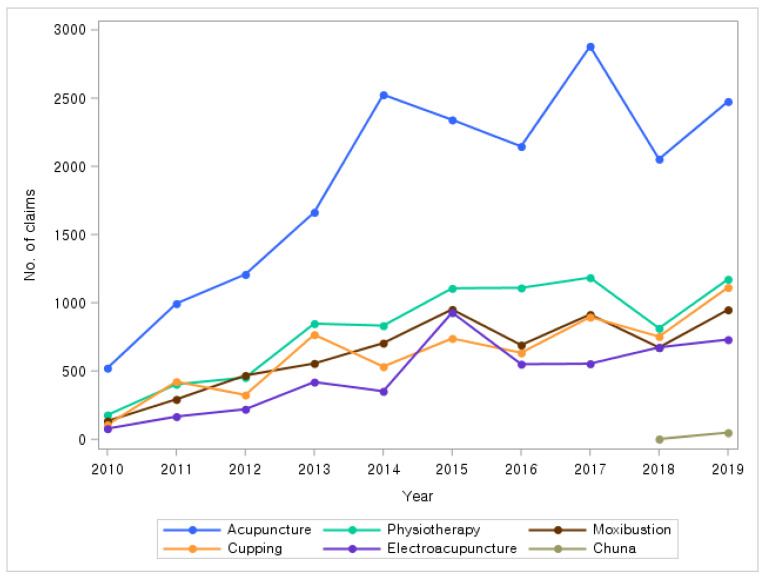
Annual trends in the number of claims for major treatment modalities of KM services. KM: Korean medicine.

**Table 1 healthcare-13-01207-t001:** Basic patient characteristics.

	Use of WM/KM Services	Total		Non-KM Users		KM Users	
Patient Characteristics		No. of Patients	Percentage (%)	No. of Patients	Percentage (%)	No. of Patients	Percentage (%)
Total		18,562	100.0	16,744	90.2	1797	9.6
Sex	Male	7290	39.3	6653	39.7	630	35.1
	Female	11,272	60.7	10,091	60.3	1167	64.9
Age (years)	<50	431	2.3	379	2.3	52	2.9
	50–59	1447	7.8	1265	7.6	179	10.0
	60–69	3783	20.4	3393	20.3	388	21.6
	≥70	12,901	69.5	11,707	69.9	1178	65.6
Payer type	NHI	16,456	88.7	14,823	88.5	1618	90.0
	Medicaid	2026	10.9	1843	11.0	177	9.9
	VHS	80	0.4	78	0.5	2	0.1

KM: Korean medicine; non-KM users: patients who used WM services only; KM users: patients who used KM services at least once; NHI: National Health Insurance; VHS: Veterans Health Service.

**Table 2 healthcare-13-01207-t002:** Basic medical usage characteristics.

		No. of Claims (Cases)	Percentage (%)
Total		149,794	100.0
Type of visit	Outpatient	119,414	79.7
	Inpatient	30,380	20.3
Medical institutions	Hospitals (tertiary/general)	83,710	55.9
	Convalescent hospitals	31,971	21.3
	Clinics	17,984	12.0
	KM hospitals	2257	1.5
	KM clinics	13,380	8.9
	Public health centers	492	0.3

KM: Korean medicine.

**Table 3 healthcare-13-01207-t003:** Primary diagnoses in claims for KM services related to PD based on the first digit.

Rank	Diagnosis Code	Name of Diagnosis	No. of Claims (Cases)	Percentage (%)
1	M	Diseases of the musculoskeletal system and connective tissue (M00–M99)	78,318	58.6
2	U	Codes for special purposes (U00–U99)	14,640	11.0
3	S	Injury, poisoning, and certain other consequences of external causes (S00–T98)	12,235	9.2
4	R	Symptoms, signs, and abnormal clinical and laboratory findings, not elsewhere classified (R00–R99)	8776	6.6
5	G	Diseases of the nervous system (G00–G99)	7053	5.3
6	K	Diseases of the digestive system (K00–K93)	3243	2.4
7	I	Diseases of the circulatory system (I00–I99)	2874	2.2
8	F	Mental and behavioral disorders (F00–F99)	2142	1.6

KM: Korean medicine; PD: Parkinson’s disease.

**Table 4 healthcare-13-01207-t004:** Analysis and classification of WM prescription details for patients with PD.

ATC Category	NHI Claims		Cost		Cost per Claim (USD)
	No. of Claims (Cases)	Percentage (%)	Total Cost (USD)	Percentage (%)	
**Anti-Parkinson drugs**					
Dopa and dopa derivatives	5365	15.2	432,592	27.2	USD 81
Dopamine agonists	3368	9.5	383,182	24.1	USD 114
Amantadine derivatives	1359	3.8	19,529	1.2	USD 14
Monoamine oxidase B inhibitors	911	2.6	127,181	8.2	USD 140
COMT inhibitor	114	0.3	13,059	0.8	USD 115
**Others**					
Alimentary tract and metabolism	4082	11.5	74,542	4.7	USD 18
Anti-dementia drugs	4025	11.4	245,852	15.4	USD 61
Cardiovascular system	3028	8.6	99,540	6.2	USD 33
Anxiolytics	1739	4.9	11,374	0.7	USD 7
Pain medications	1447	4.1	25,663	1.6	USD 18
Antidepressants	1286	3.6	23,499	1.5	USD 18
Drugs for constipation	1274	3.6	7784	0.5	USD 6
Blood and blood-forming organs	1082	3.1	21,037	1.3	USD 19
Anticonvulsants	976	2.8	4630	0.3	USD 5
Tertiary amines	860	2.4	3556	0.2	USD 4
Other nervous system medications	714	2.0	11,956	0.8	USD 17

ATC: Anatomical Therapeutic Chemical Classification System; NHI: National Health Insurance; COMT: catechol-o-methyl-transferase; PD: Parkinson’s disease; WM: Western medicine.

**Table 5 healthcare-13-01207-t005:** Analysis of basic demographic characteristics according to KM utilization frequency.

	Use of KM Services	Total		<20		≥20	
Patient Characteristics		No. of Patients	Percentage (%)	No. of Patients	Percentage (%)	No. of Patients	Percentage (%)
Total		1797	100.0	1559	86.8	238	13.2
Sex	Male	630	35.1	521	33.4	109	45.8
	Female	1167	64.9	1038	66.6	129	54.2
Age (years)	<50	52	2.9	45	2.9	7	2.9
	50–59	179	10.0	143	9.2	36	15.1
	60–69	388	21.6	308	19.8	80	33.6
	≥70	1178	65.6	1063	68.2	115	48.3

KM: Korean medicine.

**Table 6 healthcare-13-01207-t006:** Annual trends by KM utilization and anti-dementia drug prescription.

		KM ≥ 1 with Anti-Dementia Drug Prescription			
		No. of Patients	Percentage (%)	No. of Claims	Percentage (%)
Total		634	100.0	4025	100.0
Year	2010	13	2.1	72	1.8
	2011	26	4.1	156	3.9
	2012	32	5.1	187	4.7
	2013	52	8.2	345	8.6
	2014	76	12.0	480	11.9
	2015	73	11.5	415	10.3
	2016	90	14.2	622	15.5
	2017	84	13.3	590	14.7
	2018	92	14.5	621	15.4
	2019	96	15.1	537	13.3

KM: Korean medicine.

**Table 7 healthcare-13-01207-t007:** Analysis of basic demographic characteristics by KM utilization and anti-dementia drug prescription.

		KM ≥ 1 with Anti-Dementia Drug Prescription	
		No. of Patients	Percentage (%)
Total		634	100.0
Sex	Male	201	31.7
	Female	433	68.3
Age (years)	<50	1	0.2
	50–59	25	3.9
	60–69	110	17.4
	≥70	498	78.6

KM: Korean medicine.

## Data Availability

The datasets generated and/or analyzed during the current study are not publicly available due to the potential identification of individuals, but they can be obtained from the corresponding author upon reasonable request.

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
