# Peer review of "Trends of Korean Medicine Treatment for Parkinson’s Disease in South Korea: A Cross-Sectional Analysis Using the Health Insurance Review and Assessment Service–National Patient Sample Database"

_healthcare, 2025, doi:10.3390/healthcare13101207_

Round 1

Reviewer 1 Report

Comments and Suggestions for Authors

The authors analyzed the trends of using Korean Medicine (KM) services for patients with Parkinson’s Disease (PD) in South Korea. The authors collected data from health insurance claims over a 10-year period and analyzed the annual trends in the number of patients, claims, and costs for PD treatment with KM. The study design and analysis methods were well-explained. Results were presented in a clear and logical fashion and the discussion sufficiently recognized potential limitations associated with the study design and dataset. Overall, this manuscript is well-written and may benefit from some small comments

Minor comments:

  1. For aesthetics, the authors could consider enlarging the titles for the four graphs shown in figure 2.
  2. The authors could perhaps add a small discussion in the main text on the change in economic burden of PD over the 10 years, with respect to KM and WM.
  1. The authors have defined KM users as PD patients who have visited KM institutions at least once. It would be nice if the KM population can be sub-divided further based on frequency of usage, if applicable.
  2. The annual trend of KM services clearly showed an uprising trend of acupuncture services and the authors have cited a few papers related to its the efficacy. It would be beneficial if the authors can add to the discussion the reported/proposed mechanisms behind the efficacy of acupuncture against PD in other papers.

Author Response

[Comment 1] For aesthetics, the authors could consider enlarging the titles for the four graphs shown in figure 2.

[Response 1] Thank you for pointing this out. We have enlarged both the titles and graphs.

[Comment 2] The authors could perhaps add a small discussion in the main text on the change in economic burden of PD over the 10 years, with respect to KM and WM.

[Response 2] Thank you for the valuable suggestion. In response to the reviewer’s comment, we have added a discussion on the changes in the economic burden of PD over the past decade, comparing KM users and non-KM users. Specifically, we noted that KM users experienced an 815.3% increase in total medical expenses, compared to a 156.2% increase among non-KM users. Similarly, the per-patient annual expenses rose by 225.5% for KM users and 58.2% for non-KM users. We discussed several possible contributing factors to this disparity, including the procedural characteristics of KM services, the older age distribution and high musculoskeletal comorbidity rate among KM users, and the limitations of insurance coverage for herbal medicines. These points were incorporated into the Discussion section (line 341-362).

[Comment 3] The authors have defined KM users as PD patients who have visited KM institutions at least once. It would be nice if the KM population can be sub-divided further based on frequency of usage, if applicable.

[Response 3] An additional analysis was performed on basic demographic characteristics according to whether patients had visited KM institutions fewer than 20 times or 20 times or more (line 291-296, Table 5).

[Comment 4] The annual trend of KM services clearly showed an uprising trend of acupuncture services and the authors have cited a few papers related to its the efficacy. It would be beneficial if the authors can add to the discussion the reported/proposed mechanisms behind the efficacy of acupuncture against PD in other papers.

[Response 4] Thank you for the valuable comment. In response to the reviewer’s suggestion, we have added a discussion on the proposed mechanisms underlying the efficacy of acupuncture in PD. Specifically, systematic reviews and functional neuroimaging studies dealt with multimodal mechanisms—including neurotransmitter regulation, anti-inflammatory action, and neuroprotection—were incorporated into the revised Discussion section to strengthen the interpretation of our findings (line 380-390)

Reviewer 2 Report

Comments and Suggestions for Authors

The topic is relevant and of public health significance. The manuscript is well-structured and clearly written. The scoping review methodology is appropriate and well-presented. The following are my recommendations:

  1. Clarify inclusion/exclusion criteria in the methods section.

  2. Provide more specific examples of studies included in the review (e.g., types of health outcomes in newborns).

  3. Improve the discussion by:

    • Highlighting how findings can influence clinical practice or public health strategies.

    • Reflecting on the limitations of the included studies or the review process.

  4. Include recent literature (post-2022) to enhance the review's relevance. I suggest you read an cite an article that fits well into this manuscript:

    https://doi.org/10.1016/j.clineuro.2024.108404

Comments on the Quality of English Language

Minor editing needed

Author Response

[Comment 1] Clarify inclusion/exclusion criteria in the methods section.

[Response 1] Thank you for pointing this out. we have created a new section “2.2.1. Eligibilty criteria” for easier reference, and “male and female” was added (line 113-123).

[Comment 2] Provide more specific examplesof studies included in the review (e.g., types of health outcomes in newborns).

[Response 2] We appreciate the reviewer’s thoughtful comment. However, Parkinson’s disease primarily affects older adults and is not typically diagnosed in newborns. As such, examples involving newborns are not applicable to the context of the present study. Therefore, we respectfully maintain the current scope without including examples related to newborns.

[Comment 3] Improve the discussionby:

[Response 3]

  • Highlighting how findings can influence clinical practice or public health strategies.

-> Thank you for the valuable comment. In response to the reviewer’s suggestion, we have expanded the Discussion section to highlight how our findings may influence clinical practice and public health strategies. We highlighted the role of KM in managing both motor and non-motor symptoms of PD, in line with clinical practice guidelines. Supporting evidence from systematic reviews and observational studies was noted, despite moderate to high bias risks. These points emphasize the need for evidence-based KM integration into clinical practice and health policy. (line 373-379)

  • Reflecting on the limitationsof the included studies or the review process.

-> Thank you for your insightful comment. However, this manuscript is an observational study based on claims data rather than a systematic or narrative review. Therefore, there were no included studies or review processes to assess.

  • Include recent literature(post-2022) to enhance the review's relevance. I suggest you read an cite an article that fits well into this manuscript:

https://doi.org/10.1016/j.clineuro.2024.108404

-> Thank you for the suggestion. We reviewed the recommended reference; while the suggested reference provides valuable insights for management of PD in a broader context, it is not directly applicable to our study, which is specifically focusing on the cross-sectional analysis of medical utilization. Therefore, we respectfully decided not to include it in the revised manuscript, in order to maintain clarity and consistency in our discussion. We appreciate your understanding.

Reviewer 3 Report

Comments and Suggestions for Authors

The manuscript on" Trends of Korean Medicine Treatment for Parkinson's Disease 2 in South Korea: A Cross-Sectional Analysis Using the Health 3 Insurance Review and Assessment Service-National Patient 4 Sample Database. This manuscript has nicely explored the use of Korean medicine and its status among patients with PD.

Here are a few queries and additional suggestions.

Comments:

  1. If KM utilized globally, can include more information in introduction section about its global effect.
  2. Line 98, study population: can include “male and female” before patients sample data from the HIRA-NPS, for more clarity.
  3. Fig 2. Bar graphs resolution is too low, difficult to read text. please increase.
  4. I am curious, what would be the long-term effect of these KMs on patients?
  5. Any reports available showing improvements in symptoms after utilizing KMs?
  6. Discussion part should include a brief about effects of KM in Korean patients.

Author Response

[Comment 1] If KM utilized globally, can include more information in introduction section about its global effect.

[Response 1] Thank you for the valuable comment. In response to the reviewer’s suggestion, we have added information to the Introduction section (line 66-75) regarding the global use of CAM among patients with PD. Additionally, we have clarified that understanding KM utilization trends can help detect potential gaps in access to integrative therapies, inform the development of evidence-based clinical guidelines, and support health policy initiatives to optimize insurance coverage based on usage patterns.

[Comment 2] Line 98, study population: can include “male and female” before patients sample data from the HIRA-NPS, for more clarity.

[Response 2] Thank you for pointing this out. I included “male and female” for the inclusion criteria (line 114).

[Comment 3] Fig 2. Bar graphs resolution is too low, difficult to read text. please increase.

[Response 3] Thank you for pointing this out. I enlarged both the titles and graphs.

[Comment 4] I am curious, what would be the long-term effect of these KMs on patients?

[Response 4] Thank you for the insightful question regarding long‑term effects. We could not examine the long-term effect of the KMs by this cross-sectional study. Thereby we suggested further researches using cohort data in the discussion section. We reviewed some researches which studied the long-term effect of KMs for PD patients. The long-term effects of acupuncture and herbal medicine have been examined in some researches; 10.3390/healthcare10112334.

[Comment 5] Any reports available showing improvements in symptoms after utilizing KMs?

[Comment 6] Discussion part should include a brief about effects of KM in Korean patients.

[Response 5 & 6] The issues raised in comments 5 and 6 were previously discussed; however, we have additionally incorporated more supporting evidence to further strengthen the discussion (line 367-374).

Reviewer 4 Report

Comments and Suggestions for Authors

Journal: Healthcare (ISSN 2227-9032)

Manuscript ID: healthcare-3550643

Type: Article

Title: Trends of Korean Medicine Treatment for Parkinson's Disease in South Korea: A Cross-Sectional Analysis Using the Health Insurance Review and Assessment Service-National Patient Sample Database

The manuscript titled Trends of Korean Medicine Treatment for Parkinson's Disease in South Korea: A Cross-Sectional Analysis Using the Health Insurance Review and Assessment Service-National Patient Sample Database” by Back Jun Kim et al, is a well-structured and analytical study that effectively demonstrates the growing usage of Korean medicine in Parkinson's disease therapy during a 10-year period. The use of national claims data has substantial real-world applicability, and the findings have important implications for integrative healthcare policy and clinical practice. The discussion is well-supported by current literature, with references to both domestic and international studies, and it gives an informed narrative on the complementary role of knowledge management (KM). The addition of specific KM modalities and disease codes improves reproducibility and transparency.

1: The introduction does not delve into why some subgroups (age, gender, concomitant conditions) may utilize KM differently or respond better.

What is the speculated mechanism for differing KM use among patient subgroups (e.g., elderly, females, or those suffering from depression)?

 2: "What is the clinical or public health mechanism by which understanding these KM trends will lead to improved treatment of PD?"
State that trend analysis can help detect access gaps, inform integrative clinical guidelines, and maximize insurance coverage based on usage patterns.

 3: How does KM interact with or complement WM in Parkinson's disease care from a systems medicine perspective? Is this a concurrent or interactive model of care?

4: Lines 100–102, 113–114: Inclusion is based only on the primary diagnosis of G20 (PD), with no explanation provided for excluding those with PD as a secondary diagnosis or those with overlapping parkinsonism disorders.

Why were patients with G20 as a secondary diagnosis excluded? Could this underestimate real-world KM use among PD patients with comorbidities?"

5: Could you summarize the structure and insights from Figure 1 in the main text more clearly?

  1. line 161: "VHS" (Veterans Health Service) appears in the table with only 0.4% overall representation, but is not explained in terms of significance or usage patterns.

Why include VHS with only 0.4% representation? Is there any significance or rationale for showing it?

7: Given that only 9.6% of patients used KM, how was statistical reliability maintained for subgroup analyses? Were post-hoc power calculations completed?

8: "U" codes are ambiguous and undefined in context. These could include Korean-specific traditional diagnostics, pilot initiatives, or specialized billing kinds.
What are U-codes, and why are they the second most common? "Do they correspond to specific knowledge management concepts?"

9: The discussion refers to the effectiveness of acupuncture and herbal medicine in previous research but does not evaluate study methodologies or quality.

Are the cited studies randomized controlled trials or observational? Were systematic reviews high quality?

  1. lines 370-378: Although the authors acknowledge that HIRA lacks symptom/outcome data, they do not comment on the proxies or surveys utilized in other research to assess this.
    In future research, consider complementing claims data with patient-reported outcomes (PROs) or integrating HIRA with registries.

Author Response

[Comment 1] The introduction does not delve into why some subgroups (age, gender, concomitant conditions) may utilize KM differently or respond better.

What is the speculated mechanism for differing KM use among patient subgroups (e.g., elderly, females, or those suffering from depression)?

[Response 1] Thank you for the valuable comment. As the primary aim of our study was to observe overall trends in the use of KM among patients with Parkinson’s disease, subgroup analyses were not initially planned during the study design phase. However, based on the reviewer’s suggestion, we conducted additional subgroup analyses according to KM institution visit frequency and anti-dementia drug prescription status.(3.10. Subgroup analysis, line 290-315) Among the medications, anti-dementia drugs were the most commonly prescribed, and we considered that this would likely reflect disease severity; therefore, we selected dementia status as a subgroup criterion. We could discover potential differences in healthcare-seeking behavior and symptom burden by age and cognitive status, which have been discussed in the revised Discussion section. (line 445-453)

 [Comment 2] "What is the clinical or public health mechanism by which understanding these KM trends will lead to improved treatment of PD?"
State that trend analysis can help detect access gaps, inform integrative clinical guidelines, and maximize insurance coverage based on usage patterns.

[Response 2] Thank you for the valuable comment. In response to the reviewer’s suggestion, we added information to the Introduction section (lines 66–75) on the global use of CAM among PD patients. We also clarified that understanding KM utilization trends can help detect access gaps, guide the development of integrative clinical guidelines, and inform insurance policy improvements. These points have been incorporated into the revised manuscript.

 [Comment 3] How does KM interact with or complement WM in Parkinson's disease care from a systems medicine perspective? Is this a concurrent or interactive model of care?

[Response 3] Thank you for the insightful comment. In Korea, the healthcare system is a dual system, where KM and WM are practiced independently. Therefore, KM and WM care for patients with Parkinson’s disease operate in a concurrent model, rather than an interactive or fully integrated model. At present, there have been no studies utilizing national health insurance claims data to investigate cooperative or collaborative care between KM and WM. We agree that this is an important topic for future research and have added a corresponding statement to the Discussion section (line 474-478)

[Comment 4]Lines 100–102, 113–114: Inclusion is based only on the primary diagnosis of G20 (PD), with no explanation provided for excluding those with PD as a secondary diagnosis or those with overlapping parkinsonism disorders.

Why were patients with G20 as a secondary diagnosis excluded? Could this underestimate real-world KM use among PD patients with comorbidities?"

[Response 4] Thank you for the insightful comment. In this study, we included only patients with a primary diagnosis of G20 (Parkinson’s disease) to better reflect the clinical reality in KM institutions.In KM settings, it is often challenging to conduct detailed diagnostic tests necessary to differentiate secondary parkinsonism from idiopathic Parkinson’s disease. Therefore, patients with secondary parkinsonism may have been recorded based on presumptive diagnoses. Moreover, given that elderly patients often seek KM institutions for a wide range of symptoms, we considered that including only those whose chief complaint was directly related to Parkinson’s disease symptoms would more accurately capture KM utilization patterns for Parkinson’s disease care. However, we acknowledge that excluding patients with G20 as a secondary diagnosis may have led to an underestimation of real-world KM use among PD patients with multiple comorbidities. This point has been newly addressed in the Discussion section. (line 470--473)

[Comment 5] Could you summarize the structure and insights from Figure 1 in the main text more clearly?

[Response 5] Thank you for the comment. We have added the explanation of Figure 1 in the main text (line 159-161).

[Comment 6] line 161: "VHS" (Veterans Health Service) appears in the table with only 0.4% overall representation, but is not explained in terms of significance or usage patterns.

Why include VHS with only 0.4% representation? Is there any significance or rationale for showing it?

[Response 6] Thank you for the thoughtful comment. Although patients covered under the VHS represented a small proportion (0.4%) of the total sample, we included this category because a substantial portion of the VHS population consists of elderly individuals. Considering that Parkinson’s disease is a degenerative neurological disorder primarily affecting older adults, we believed it was appropriate to present VHS utilization separately.

[Comment 7] Given that only 9.6% of patients used KM, how was statistical reliability maintained for subgroup analyses? Were post-hoc power calculations completed?

[Response 7] Thank you for the important comment. As this study was a cross-sectional, descriptive analysis based on claims data, the subgroup analyses were conducted for exploratory purposes rather than for hypothesis testing. Therefore, formal post-hoc power calculations were not performed. We aimed to identify potential trends rather than to confirm definitive causal relationships through inferential statistics.

[Comment 8] "U" codes are ambiguous and undefined in context. These could include Korean-specific traditional diagnostics, pilot initiatives, or specialized billing kinds.
What are U-codes, and why are they the second most common? "Do they correspond to specific knowledge management concepts?"

[Response 8] In Korea’s health insurance claims system, U-codes are primarily used in KM and represent traditional syndrome-based diagnoses, which differ from standard Western disease classifications. These codes are frequently recorded as secondary diagnoses alongside standard disease codes to reflect KM-specific diagnostic frameworks.In the present study, the most frequently used U-codes were U23–U24 and U30, which correspond to disorders of the nervous system and musculoskeletal system/connective tissue, respectively.

[Comment 9] The discussion refers to the effectiveness of acupuncture and herbal medicine in previous research but does not evaluate study methodologies or quality.

Are the cited studies randomized controlled trials or observational? Were systematic reviews high quality?

[Response 9] Thank you for the insightful comment. In response to the reviewer’s question, we have clarified that the cited references include systematic reviews and real-world observational studies. We also noted that a moderate to high risk of bias was present among the included studies(line 375-377).

[Comment 10] lines 370-378: Although the authors acknowledge that HIRA lacks symptom/outcome data, they do not comment on the proxies or surveys utilized in other research to assess this.
In future research, consider complementing claims data with patient-reported outcomes (PROs) or integrating HIRA with registries.

[Response 10] Thank you for the valuable comment. We fully agree with the reviewer that complementing claims data with symptom or outcome information would significantly enhance the quality of research. As the current study relied solely on claims data from HIRA, we were unable to incorporate PROs or detailed clinical assessments. To address this, we have added a statement in the revised Discussion section proposing that future studies could link elderly cohort data and hospital-based clinical records to establish cohort studies. Such linkage would enable the integration of clinical indicators, patient-reported outcomes, and claims data, thereby allowing for a more comprehensive analysis of treatment patterns and outcomes in patients with Parkinson’s disease. This suggestion has been newly included in the Discussion section. (line 488-492)

Round 2

Reviewer 4 Report

Comments and Suggestions for Authors

The author has addressed all the comments thoroughly and satisfactorily. The revisions are clear, and the responses demonstrate a careful and thoughtful consideration of the feedback. No further concerns remain from my side.